# Pharmacological Strategies for Suicide Prevention Based on the Social Pain Model: A Scoping Review

**Ravi Philip Rajkumar** 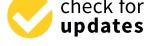

Department of Psychiatry, Jawaharlal Institute of Postgraduate Medical Education and Research (JIPMER), Puducherry 605 006, India; jd0422@jipmer.ac.in; Tel.: +91-413-2296280

**Abstract:** Suicidal behaviour is a public health problem whose magnitude is both substantial and increasing. Since many individuals seek medical treatment following a suicide attempt, strategies aimed at reducing further attempts in this population are a valid and feasible secondary prevention approach. An evaluation of the available evidence suggests that existing treatment approaches have a limited efficacy in this setting, highlighting the need for innovative approaches to suicide prevention. Existing research on the neurobiology of social pain has highlighted the importance of this phenomenon as a risk factor for suicide, and has also yielded several attractive targets for pharmacological strategies that could reduce suicidality in patients with suicidal ideation or a recent attempt. In this paper, the evidence related to these targets is synthesized and critically evaluated. The way in which social pain is related to the "anti-suicidal" properties of recently approved treatments, such as ketamine and psilocybin, is examined. Such strategies may be effective for the short-term reduction in suicidal ideation and behaviour, particularly in cases where social pain is identified as a contributory factor. These pharmacological approaches may be effective regardless of the presence or absence of a specific psychiatric diagnosis, but they require careful evaluation.

**Keywords:** suicide; social pain; psychache; endogenous opioid system; oxytocin; serotonin; endocannabinoids; buprenorphine; psilocybin; ketamine



## 1. Introduction

Suicide is one of the leading global causes of premature mortality. It is estimated that around 750,000–800,000 people lose their lives to suicide each year, and the suicide rates appear to be rising in several low- and middle-income countries [1,2]. Suicide is a complex phenomenon, representing the result of non-linear interactions between an innate genetic or developmental vulnerability [3–5], current life stressors or events [6], and individual or social protective factors [7]. Suicide attempts are preceded by suicidal ideation and planning, but suicidal ideation is relatively common in the general population, and many individuals with such ideas do not progress to making a suicide attempt [8–10]. A suicide attempt is one of the most robust predictors of future suicide risk; therefore, the provision of interventions in the aftermath of an attempt is one of the key secondary suicide prevention strategies [11]. Several psychological interventions to reduce suicidal ideation and behaviour have been evaluated in controlled trials and a critical evaluation of these strategies found that the benefits from such interventions are modest, and that there is no evidence for the superiority of one approach over another [12,13]. Likewise, in view of the association between mental illness and suicide, several pharmacological agents have been assessed for "anti-suicide" effects. There is no significant evidence that the existing pharmacotherapies have robust and replicated effects on suicide risk, with the exception of two drugs—lithium in patients with bipolar disorder, and clozapine in patients with treatment-resistant schizophrenia. Neither of these drugs is useful as a suicide prevention strategy in the absence of these specific diagnoses, and both are associated with significant adverse effects and a narrow therapeutic index [14]. Furthermore, there is

evidence that some pharmacological agents may slightly but significantly increase the risk of suicide, particularly in youth [15]. These findings suggest that there is an urgent need for approaches to suicide prevention, particularly following a suicide attempt, which have a greater efficacy and a better risk-benefit ratio than existing pharmacological or psychological therapies.

In recent years, much has been learned about the neuroanatomical, neurophysiological and biochemical correlates of suicidal ideation and behaviour. There is little evidence for gross structural abnormalities in the brains of individuals attempting suicide. Instead, functional imaging studies have revealed evidence of reduced activity in the prefrontal cortices, left insula and right putamen, possibly related to altered serotonergic transmission, and increased activity in the superior and middle temporal gyri and right occipital cortex [16,17]. Biochemical studies have revealed evidence of alterations related to monoaminergic transmission [18], hypothalamic–pituitary–adrenal (HPA) axis activity [19], immune-inflammatory pathways [20], and neural plasticity [21]. Recent genome-wide analyses of individuals who have made a suicide attempt are consistent with these findings, suggesting that several biological pathways are related to suicidality, including monoaminergic, glutamatergic and peptidergic neurotransmission, the HPA axis and circadian rhythms [22]. Likewise, psychological factors such as a decision-making ability, executive functioning, neuroticism, impulsivity and aggression have been identified as candidate endophenotypes for suicidal behaviour [23]. Though each of these findings is valuable and could serve as a potential lead for the development of pharmacological or psychological suicide prevention strategies, it is unclear to what extent they apply to individual cases encountered in a "real-world" setting such as a clinic, emergency room or community treatment centre. It is also unclear to what extent these individual pathways are inter-related. The existing literature highlights the need for an overarching theory—or at least a hypothesis—that could link these findings and highlight discrete targets for suicide prevention strategies that can be tested in controlled trials.

In 1993, Edwin Shneidman suggested that, regardless of the role of specific triggering factors, the ultimate reason for suicide was an attempt to put an end to "unbearable mental pain", which he termed psychache [24]. Subsequent research has confirmed Shneidman's hypothesis [25], and has found that mental or psychological pain may be a stronger predictor of suicide than other known risk factors, such as the presence of depression [26]. Recent neurobiological research has deepened our understanding of the phenomenon of psychological or mental pain by identifying common neural pathways and mechanisms that underlie the response to both physical pain and to the distressing response to conflict or disruption in social relationships, which is known as "social pain" [27]. From an empirical perspective, the conditions associated with social pain are frequently cited as motives for suicide. These include conflicts between family members (such as parent–child conflict), marital conflict, infidelity by a spouse, a lack of close friends, a perceived lack of support or affection from significant others, loneliness, bullying (including "cyber-bullying"), and social ostracism [28–32]. These factors appear to play a significant role regardless of the age or geographical location of the populations being studied, suggesting that they may all act through the final common pathway of causing intolerable social pain. The available evidence suggests that social pain is, in fact, a key mediator between an exposure to such situations and suicidal behaviour [33].

From a neuroanatomical perspective, social pain appears to be strongly correlated with altered patterns of activation in distinct areas of the anterior cingulate cortex (ACC) [34]. Other components of the "physical" pain pathway that have also been implicated in the experience social pain include the thalamus, sensory cortex, and periaqueductal gray area of the midbrain [27]. Neurochemically, social pain appears to be most strongly linked to alterations in the activity of the endogenous opioid system, particularly to changes in the activation of μ-opioid receptors [35–37]. This pathway is known to be crucially involved in the modulation of physical pain [38], supporting the hypothesis of an overlap between the two. Other neurotransmitters that have been linked to the perception of social pain

include serotonin [39], the neuropeptide oxytocin [40], and endogenous cannabinoids [41]. Pro-inflammatory cytokines, which are known to increase sensitivity to physical pain, have also been associated with increased neural activation in key brain regions when exposed to an experimental situation that induces social pain; this suggests a certain degree of cross-talk between the immune-inflammatory and social pain pathways [42]. Though this hypothesis assumes an overlap between the neural substrates of physical and social pain, it does not require a perfect correspondence between the two. Functional brain imaging studies have found that, though both phenomena involve alterations in the functioning of common brain regions, the patterns of altered activation in these regions can reliably distinguish between the two [43].

Based on this contemporary understanding of social pain, John Gunn has extended Shneidman's hypothesis through his proposal of the social pain model of suicide [44]. According to this model, experiences such as rejection or exclusion in a social or interpersonal setting result in social pain. In vulnerable individuals, social pain is associated with "suicidogenic cognitions", such as ideas of hopelessness, of being a burden to others, or of feeling trapped in a situation from which one cannot escape. Social pain can also be associated with "negative social cognitions" which, while not directly suicidogenic, can reinforce suicidogenic cognitions; these may include ideas of oneself as inferior, inept and being unable to sustain social relationships. Social pain, suicidogenic cognitions and negative social cognitions may then create a positive feedback loop, which—if left uninterrupted—can lead to an increase in social pain to an intolerable level, which triggers suicidal behaviour.

Though Gunn's formulation is based on a consideration of psychological intermediate variables, it is supported by neuroimaging studies of individuals with a recent or past suicide attempt. When exposed to an experimental state of social exclusion, these individuals were found to have reduced activation in specific cortical regions and decreased oxytocin levels; alterations in regional brain activity were correlated with levels of cytokines such as interleukin-1 beta (IL-1β) and interleukin-2 (IL-2) [45]. Similarly, a study of individuals with low social integration and acute suicidal ideation found evidence of increased activity in the dorsal ACC and anterior insula, which appear to play a role in the experience of social pain [46], while a study of women with a past history of suicidal behaviour showed evidence of reduced activity in the left insula and supramarginal gyrus, in contrast with healthy controls, when experiencing social exclusion in an experimental setting [47]. Though these findings require replication, they provide a support for alterations in the neural substrates of social pain perception in relation to suicidal ideation or behaviour.

It has been suggested that the endogenous opioid system may play a central role in the link between an exposure to adverse social experiences, social pain and suicide [36]. There is evidence from animal and human research that the distress caused by separation or social exclusion is related to altered signalling at the μ-opioid receptor. The extent of this alteration may be positively correlated with the severity of social pain experienced by a given person, and it may vary in relation to genetic polymorphisms of the μ-opioid receptor gene (*OPRM1*) [48,49]. If this pain crosses a certain threshold, suicidal behaviour is more likely to occur according to the social pain model. In line with this proposal, there is evidence that a specific functional polymorphism of the *OPRM1* gene is associated with an increased risk of suicide [50,51].

In summary, the social pain model of suicide postulates that exposure to certain adverse experiences triggers social pain, which can result in negative cognitions and form a self-reinforcing process, resulting in suicidal behaviour; this process may be mediated at a neural level by alterations in opioidergic transmission. While this model may not account for all suicide attempts, it may be of particular relevance to those individuals whose suicide attempts occur in the context of events that trigger social pain [52]. In such cases, pharmacological or psychological therapies aimed at altering social pain perception or sensitivity may represent valid suicide prevention strategies.

The social pain model and its functional correlates are summarized in Figure 1.

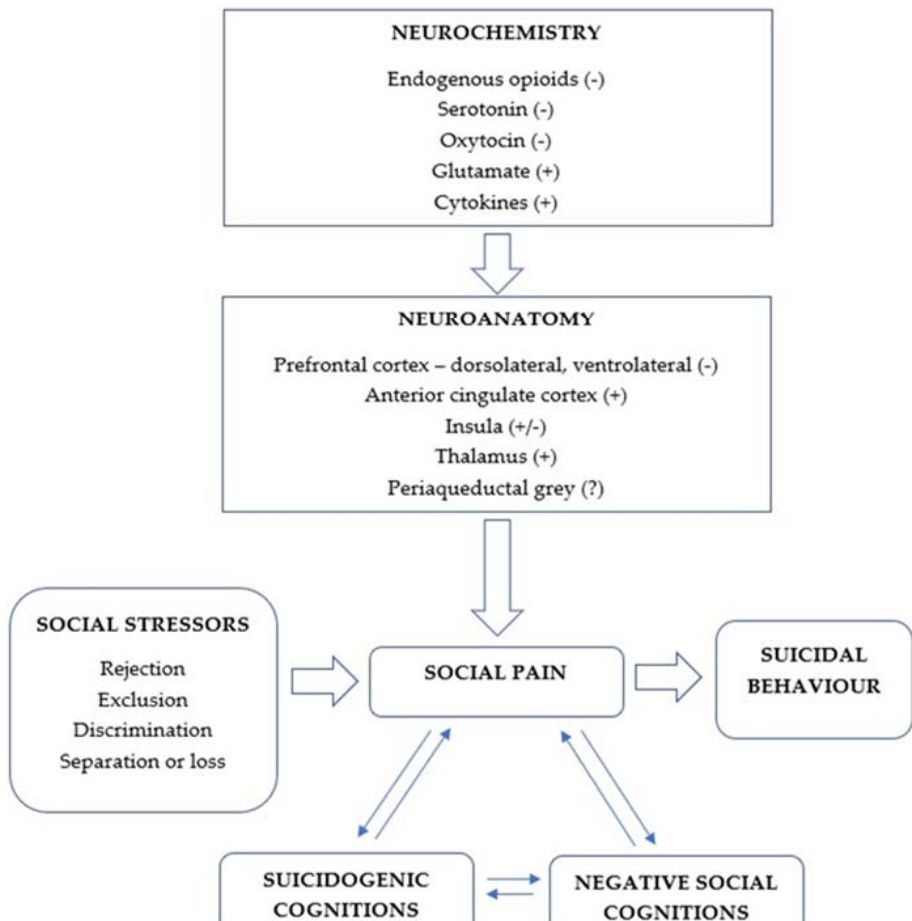

**Figure 1.** The social pain model of suicide and its biological substrates. (+) indicates a transmitter or region that may be associated with an increase in social pain; (−) indicates a transmitter or region that may attenuate social pain; (+/−) indicates evidence of mixed effects; (?) indicates an uncertain effect.

The objective of the current review is to evaluate the available literature on the association between social pain and suicide, and on the neurobiological mechanisms underlying this association, with a specific focus on possible targets for pharmacological intervention.

## 2. Materials and Methods

The current paper is a scoping review of the links between social pain and suicide, with a specific focus on potential targets for pharmacological interventions. This review was carried out in accordance with the PRISMA Extension for Scoping Reviews (PRISMA-ScR) guideline and checklist [53]. For the purpose of this review, a literature search of the PubMed/MEDLINE, Scopus and ScienceDirect literature databases was conducted, using the following search terms:

- ("social pain", "social pain model", "social pain hypothesis", "social rejection" OR "social stress") along with ("suicide", "suicide attempt", "suicide attempter", "suicidal ideation", "suicidal behaviour", "suicide risk", "suicide prevention" or "suicidality"). This search yielded a total of 287 citations (PubMed—140, Scopus—19, ScienceDirect—128)
- ("endogenous opioid", "opioid receptor", "serotonin", "serotonergic", "cannabinoid", "endocannabinoid", "oxytocin", "cytokine" OR "interleukin") along with ("suicide prevention", "suicide attempt", "suicide attempter", "suicidal ideation", "suicidal ideator", "suicide risk" or "suicidality") along with ("social pain", "social stress" OR "social rejection"). The purpose of this second search was to identify studies examining the known neural substrates of social pain in relation to suicidality. This search yielded a total of 279 citations (PubMed—133, Scopus—87, ScienceDirect—59).

Thus, a total of 566 citations were screened for possible inclusion in the scoping review. After the exclusion of duplicate citations (n = 205), the remaining 361 citations were evaluated for inclusion. Based on an examination of the title and abstract, 163 citations were excluded as being unrelated to the subject of this review. The complete texts of the remaining 198 articles were examined for suitability. Articles were included only if they reported original findings, or syntheses of original research (systematic reviews and meta-analyses), examining the links between social pain and suicide, with a specific focus on neurobiological mechanisms or pharmacological therapies. Studies of psychosocial correlates of social pain, psychotherapy trials, general review articles on suicide or social pain, and commentaries or editorials were excluded, as were papers that were unrelated to the subject of this review. Through this process, 52 papers were selected for inclusion. As a final step, the reference lists of the included articles were searched for relevant articles that might have been missed through the above search; five further papers were retrieved through this method. A total of 57 papers were included in the final review [39,41,42,54–107]. The search strategy is summarized in the PRISMA-ScR flow diagram below (Figure 2).

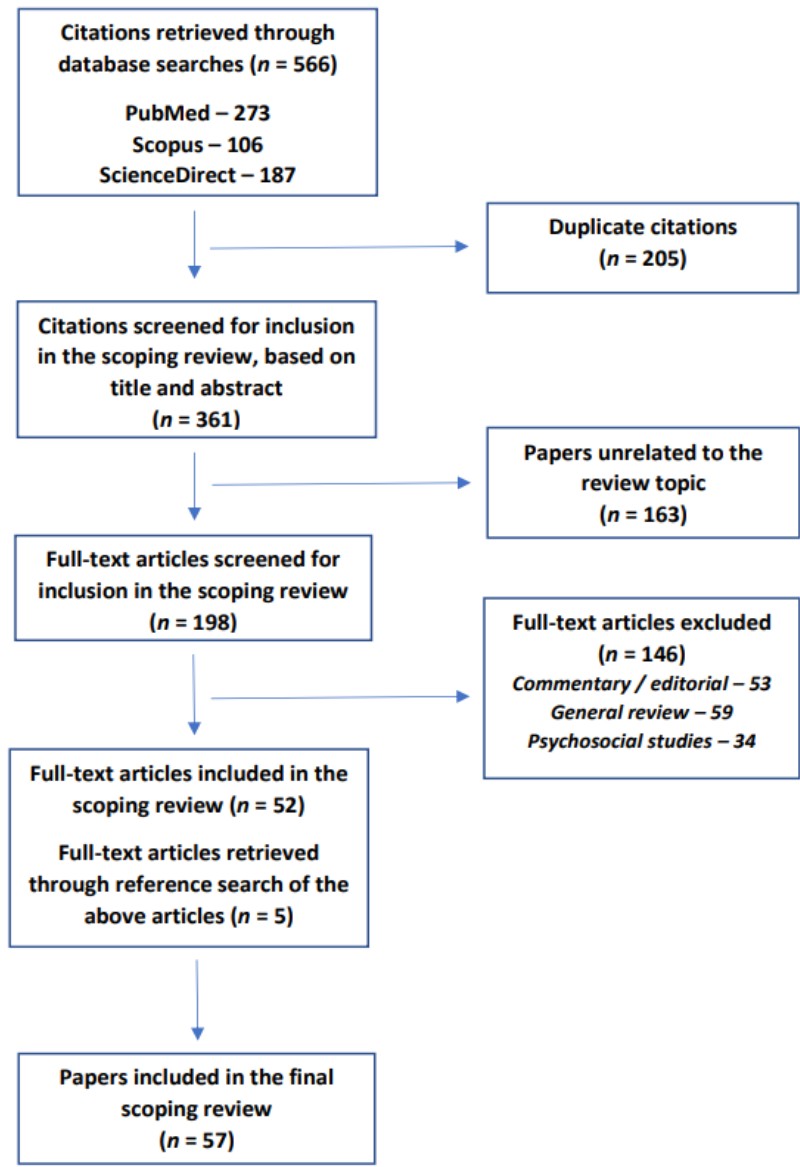

**Figure 2.** PRISMA-ScR flow diagram depicting the selection of articles for inclusion in this review.

## 3. Results

The 57 papers included in this review are summarized in detail in Table 1.

**Table 1.** Detailed description of the papers included in the current scoping review.

| Publication | Publication Type | Study Population or Data Source | Key Findings |
|---|---|---|---|
| **Agents acting at opioid receptors** | | | |
| Ahmadi et al., 2018 [54] | Randomized controlled trial | 51 male patients with comorbid major depression, opioid dependence and suicidal ideation | Single-dose buprenorphine (32, 64 or 96 mg) associated with a significant reduction in suicidal ideation; effect sustained at 2 weeks. |
| Benhamou et al., 2021 [55] | Case report | Male patient with bipolar disorder, opioid use disorder and recent suicide attempt | Buprenorphine therapy associated with rapid improvement in depression and suicidal ideation. |
| Cameron et al., 2021 [56] | Systematic review | Human and animal studies on the anti-suicidal effects of opioid receptor modulation | Based on analysis of literature, μ-receptor agonism unlikely to be the sole mechanism of anti-suicidal effects; κ-receptor antagonism probably of equal or greater importance. |
| Coplan et al., 2017 [57] | Retrospective cohort study | United States National Poison Data System | Buprenorphine associated with a significantly lower risk of abuse and suicide intent compared to all other opioid analgesics. |
| Gibbs et al., 2020 [58] | Retrospective chart review | Case records of 6 adult patients with depression, chronic pain and opioid use disorder with suicidal risk | 4 of 6 patients showed reduced suicidal ideation with off-label buprenorphine. |
| Ma et al., 2022 [59] | Animal study | Rat model of physical and affective pain | Activation of δ-opioid receptors by a synthetic opioid agonist led to a reduction in affective pain. |
| Striebel and Kalapatapu 2014 [60] | Case report | Female patient with treatment resistant depression, opioid use disorder, recent suicide attempt | Sublingual buprenorphine (16 mg/day) associated with rapid reduction in suicidal ideation and no further attempts; effect sustained at 3 month follow-up. |
| Thase et al., 2020 [61] | Open-label trial | 769 patients with depression | Combined buprenorphine and samidorphan (2 mg/2 mg/day) not associated with any significant change in suicidal ideation, despite sustained improvements in depression at 6 and 12 months. |
| Yovell et al., 2016 [62] | Randomized controlled trial | 88 patients with severe suicidal ideation and no history of substance abuse | Buprenorphine (mean dose 0.44 mg/day) associated with a significant decrease in suicidal ideation at 2 and 4 weeks compared to placebo. |
| **Agents acting at serotonin receptors** | | | |
| Argento et al., 2017 [63] | Community-based cohort study | 766 female sex workers in an urban area | Lifetime psychedelic use associated with a 60% reduction in hazard ratio for suicidality. |
| Davis et al., 2021 [64] | Randomized controlled trial | 24 patients with depression | 2 doses of oral psilocybin (20–30 mg/70 kg) in combination with 11 h of psychotherapy did not lead to a significant decrease in suicidal ideation compared to waiting-list controls. |
| Ghavidel-Parsa et al., 2022 [65] | Randomized controlled trial | 44 patients with fibromyalgia | Duloxetine (30–60 mg/day) did not lead to significant decreases in social pain despite improvement in physical pain. |
| Hendricks et al., 2015a [66] | Secondary analysis of survey data | United States National Survey on Drug Use and Health, 2008–2012; data on 191,832 participants | Lifetime psychedelic use associated with significantly lower psychological distress, suicidal ideation, planning or attempts. |
| Hendricks et al., 2015b [67] | Secondary analysis of survey data | United States National Survey on Drug Use and Health, 2008–2012; data on 191,832 participants | Psilocybin use associated with significantly lower levels of psychological distress and suicidal ideation, planning or attempts. |
| Jones and Nock 2022 [68] | Secondary analysis of survey data | United States National Survey on Drug Use and Health, 2008–2019; data on 484,732 participants | Psilocybin and MDMA use both associated with significantly lower levels of psychological distress and suicidal ideation. |

**Table 1.** *Cont.*

| Publication | Publication Type | Study Population or Data Source | Key Findings |
|---|---|---|---|
| Mitchell et al., 2021 [69] | Randomized controlled trial | 90 patients with severe PTSD | No significant difference in suicidality between MDMA and placebo. |
| Mithoefer et al., 2018 [70] | Randomized controlled trial | 26 patients with PTSD | MDMA (30, 75 or 125 mg) plus psychotherapy associated with a significant decrease in suicidality post-treatment; however, transient initial increases in suicidality were observed with the 30 mg and 125 mg doses. |
| Preller et al., 2016 [39] | Randomized cross-over trial | 21 healthy participants subjected to a social exclusion task | Administration of the 5HT2A/1A agonist psilocybin (0.215 mg/kg) associated with reduced feelings of social pain. |
| Ross et al., 2021 [71] | Randomized controlled trial | 11 patients with advanced cancer and suicidal ideation | Single-dose psilocybin (0.3 mg/kg) in combination with psychotherapy resulted in significant decreases in suicidal ideation compared to placebo + psychotherapy; effect sustained at 6 months. |
| Yang et al., 2022 [72] | Secondary analysis of survey data | United States National Survey on Drug Use and Health, 2015–2020; data on 241,675 participants | MDMA use associated with significantly lower levels of psychological distress and suicidal ideation. |
| **Agents acting at glutamate receptors** | | | |
| Can et al., 2021 [73] | Open-label trial | 32 patients with chronic suicidal ideation | Oral ketamine (0.5–3 mg/kg) associated with significant reduction in suicidal ideation over 6 weeks; effect sustained up to 4 weeks after stopping medication |
| Canuso et al., 2018 [74] | Randomized controlled trial | 68 patients with depression and imminent suicide risk | Intranasal esketamine (84 mg twice weekly) superior to placebo in reducing depressive symptoms and suicidal ideation, but not clinician-assessed suicide risk. |
| Chen et al., 2018 [75] | Post-mortem study | Brain tissue of 26 suicidal deaths and 14 controls | Agmatine levels significantly reduced in the cortex of suicides, independent of a diagnosis of depression. |
| De Gioannis and De Leo 2014 [76] | Case report | Male patient with treatment-resistant depression and chronic suicidality | Oral ketamine (0.5 mg/kg) associated with a rapid improvement in suicidal ideation. |
| Gallay et al., 2021 [77] | Secondary analysis of imaging data from open-label trial | 30 patients with chronic suicidal ideation | Oral ketamine (0.5–3 mg/kg) associated with increased grey matter volume in periaqueductal gray, bilateral basal ganglia and thalami. |
| Fu et al., 2020 [78] | Randomized controlled trial | 226 patients with depression and active suicidal ideation | Intranasal esketamine (84 mg twice weekly) superior to placebo for depressive symptoms, but not suicidality. |
| Garcia-Pardo et al., 2019 [79] | Animal study | Mouse model of social defeat | Blockade of NMDA and AMPA glutamate receptors attenuated the effects of social defeat. |
| Ghavidel-Parsa et al., 2022 [65] | Randomized controlled trial | 27 patients with fibromyalgia | Pregabalin (75–150 mg/day) did not lead to significant decreases in social pain despite improvement in physical pain. |
| Ionescu et al., 2021 [80] | Randomized controlled trial | 227 patients with depression and active suicidal ideation and intent | Intranasal esketamine (84 mg twice weekly) superior to placebo for depression, but not suicidality. |
| Irwin et al., 2013 [81] | Open-label trial | 14 patients in hospice care with depression | Oral ketamine (0.5 mg/kg) associated with significant improvements in depression over 4 weeks, but no change in suicidal ideation. |
| Kim et al., 2022 [82] | Animal study | Mouse model of chronic social stress | Expression of mGlu5 receptors in specific cortical areas associated with resilience to social stress. |
| Lewis et al., 2020 [83] | Observational study | 24 adolescents with depression, 11 with and 13 without suicidal ideation | Suicidal ideation associated with higher glutamate levels in anterior cingulate cortex; association between glutamate and suicidal ideation mediated by psychological pain. |

**Table 1.** *Cont.*

| Publication | Publication Type | Study Population or Data Source | Key Findings |
|---|---|---|---|
| Vieira et al., 2021 [84] | Randomized controlled trial | 59 patients with treatment-resistant depression and suicidal ideation | Single infusions of ketamine (0.5 mg/kg) and esketamine (0.25 mg/kg) equally effective in reducing suicidal ideation; effect sustained up to 7 days post-infusion. |
| Williams et al., 2019 [85] | Randomized cross-over trial | 12 patients with depression, 11 of whom had active suicidal ideation | Naltrexone 50 mg significantly attenuated the effect of ketamine 0.5 mg/kg on suicidal ideation. |
| Witt et al., 2020 [86] | Meta-analysis of randomized controlled trials | Data on 15 trials involving 572 patients with mood or anxiety disorders | Ketamine 0.5 mg/kg associated with significant reduction in suicidal ideation persisting up to 3 days; no data on suicide attempts. |
| **Agents acting at oxytocin receptors** | | | |
| Auer et al., 2015 [87] | Observational study | 94 healthy volunteers exposed to a social stress task | Oxytocin receptor gene OXTR rs53576 polymorphism interacts with rejection sensitivity to influence cortisol response to social stress. |
| Chu et al., 2020 [88] | Observational study | 31 suicide attempters, 21 controls with depression but no suicide attempt, 48 healthy controls | Oxytocin levels decreased following exposure to an experimental task simulating social exclusion in suicide attempters, but not in the other groups. |
| De Cagna et al., 2019 [89] | Systematic review of randomized controlled trials | 15 clinical trials of intranasal oxytocin in depression or anxiety disorders | No significant effect of oxytocin on core symptoms of anxiety or depression. |
| Fang et al., 2014 [90] | Randomized controlled trials | 60 male volunteers with no history of psychiatric illness or active suicidal ideation | No significant direct effect of oxytocin on reactions to an experimental social exclusion task; possible interaction with attachment style, with a favourable effect observed only in those with low attachment avoidance. |
| Henningsson et al., 2021 [91] | Randomized controlled trial | 100 healthy volunteers (50 pairs of men and women) | No significant direct effect of oxytocin on reactions to an experimental social exclusion task; possible interaction between gender and oxytocin, with a favourable effect observed only in women. |
| Jahangard et al., 2020 [92] | Observational study | 16 suicide survivors; 16 recent (<12 weeks) suicide attempters; 16 healthy controls | Serum oxytocin significantly lower in suicide survivors than controls; oxytocin level negatively correlated with suicidal ideation. |
| Jokinen et al., 2012 [93] | Observational study | 28 suicide attempters and 19 healthy volunteers | Cerebrospinal fluid oxytocin lower in attempters than controls; oxytocin level negatively correlated with suicide intent in attempters. |
| Krimberg et al., 2022 [94] | Meta-analysis of animal studies | 12 studies examining the relationship between oxytocin and social isolation in rodents | Oxytocin administration reverses the behavioural problems (aggression, anxiety, overactivity, reduced social interaction) caused by social isolation. |
| Lebowitz et al., 2019 [95] | Observational study | 168 adolescents with anxiety disorders | Negative interactions with peers were associated with suicidal ideation only in adolescents with high oxytocin levels. |
| Okumura et al., 2019 [96] | Animal study | Rat model of visceral pain | Centrally administered oxytocin had analgesic effects which were blocked by centrally administered naloxone, indicating that this effect is mediated through the central opioid system. |
| Radke et al., 2021 [97] | Randomized controlled trial | 43 healthy women randomized to oxytocin (n = 22) or placebo (n = 21) | No significant effect of oxytocin on behavioural or neuroimaging responses to a simulation of online social rejection. |
| Zhang et al., 2021 [98] | Randomized controlled trial | 61 healthy volunteers randomized to oxytocin (n = 30) or placebo (n = 31) exposed to an experimental task simulating romantic rejection | Oxytocin associated with a reduction in social pain compared to placebo, based on self-report and analysis of EEG data. |

**Table 1.** *Cont.*

| Publication | Publication Type | Study Population or Data Source | Key Findings |
|---|---|---|---|
| **Other molecular pathways of interest** | | | |
| Chiurliza and Joiner 2018 [99] | Randomized controlled trial | 43 healthy volunteers randomized to acetaminophen or no treatment | No significant effect of acetaminophen (1000 mg, single dose) on measures of capacity for suicide. |
| Dewall et al., 2010 [100] | Randomized controlled trial | 62 healthy volunteers randomized to acetaminophen (n = 30) or placebo (n = 32) | Acetaminophen 1000 mg/day associated with reduced social pain over 21 days; this effect was associated with reduced activation of the dorsal anterior cingulate cortex and anterior insula. |
| Eisenberger et al., 2009 [42] | Randomized controlled trial | 36 healthy volunteers randomized to endotoxin (n = 20) or placebo (n = 16) | Endotoxin associated with increased levels of IL-6, depression, and social pain compared to placebo. |
| Hungund et al., 2004 [101] | Post-mortem study | Brain tissue from 10 completed suicides and 10 normal controls | Increased $CB_1$ receptor density and receptor-stimulated binding observed in the dorsolateral prefrontal cortex in completed suicides. |
| Mato et al., 2018 [102] | Post-mortem study | Brain tissue from 12 patients with depression and completed suicide and 12 matched controls | Increased functional coupling of $CB_1$ receptors in the prefrontal cortex of cases, but not controls. |
| Muscatell et al., 2015 [103] | Observational study | 31 healthy female volunteers exposed to a social stress task | Social rejection associated with elevated IL-6 and increased amygdala activity; greater increase in IL-6 correlated with amygdala activation. |
| Ostadhadi et al., 2016 [104] | Animal study | Mouse model of depression | Administration of a $CB_1$ receptor inverse agonist associated with antidepressant effects; this effect was potentiated by administration of naltrexone. |
| Raison et al., 2013 [105] | Randomized controlled trial | 60 patients with depression randomized to infliximab (n = 30) or placebo (n = 30) | In patients with elevated baseline hs-CRP, infliximab was superior to placebo in reducing depressive symptoms and suicidality. |
| Schneider et al., 2016 [41] | Animal study | Rat model of social rejection | Administration of rimonabant attenuated the effects of social rejection on behaviour and social pain. |
| Slavich et al., 2010 [106] | Observational study | 124 healthy volunteers exposed to a social stress task | Social rejection associated with elevations in IL-6 and sTNFalphaRII; sTNFalphaRII levels correlated with increased activity in dorsal anterior cingulate cortex and anterior insula. |
| Slavich et al., 2019 [107] | Randomized controlled trial | 42 healthy volunteers randomized to acetaminophen (n = 15), pill placebo (n = 14) or no treatment (n = 13) | Acetaminophen (1000 mg/day) superior to control groups in reducing social pain, but only in subjects with high levels of forgiveness. |

Abbreviations: $5\text{-HT}_{2A/1A}$, serotonin type 2A/1A receptors; AMPA, α-amino-3-hydroxy-5-methyl-4-isoxazolepropionic acid; $CB_1$, cannabinoid type 1 receptor; EEG, electroencephalogram; hs-CRP, high-sensitivity C-reactive protein; IL-6, interleukin-6; MDMA, 3,4-methylenedioxymethamphetamine; mGlu5, metabotropic glutamate receptor type 5; NMDA, *N*-methyl d-aspartate; OXTR, oxytocin receptor gene; PTSD, post-traumatic stress disorder; sTNFalphaRII, soluble tumor necrosis factor alpha receptor type II.

Based on the content of the retrieved articles, the key findings of this review have been organized according to potential neurobiological targets or mechanisms, using the same order followed in Table 1.

(a) Opioid receptor agonists and antagonists

Endogenous opioid peptide transmission, and signalling at the μ-opioid receptor in particular, appears to play a central role in the perception of social pain; therefore, it is possible that cautious pharmacological manipulation of this receptor could reduce the severity of social pain, and thereby reduce suicidal cognitions and behaviours [36–38,48–51]. The possible benefits of this approach must be weighed against the significant evidence for an increased risk in suicide related to high-dose opioid therapy or to its discontinuation. There is some evidence that buprenorphine, a partial μ-agonist with a high receptor affinity, may be associated with a lower risk of abuse and suicide than other opioid analgesics [57]; thus, it is possible that therapy with buprenorphine—but not other commonly used opioids—could be effective in the reduction in suicidality through its effects on social pain.

Several case reports suggest that buprenorphine may have a beneficial short-term effect on suicidal ideation and behaviour [55,56,58]. To evaluate this possibility with more rigour, Yovell et al. conducted a multi-centre, placebo-controlled trial of low-dose buprenorphine (0.1–0.8 mg/day; the standard analgesic dose is 4–24 mg/day) in patients with significant suicidal ideation and no past or current history of substance abuse [62]. It was found that buprenorphine was significantly superior to placebo in reducing suicidal ideation after 2 and 4 weeks, and was also superior in reducing self-reported mental pain. These results were not influenced by concurrent antidepressant treatment, and buprenorphine did not cause a significant reduction in the symptom scores for depression, suggesting that its suicide-specific effect was independent of any antidepressant property. Though these results are promising, they should be interpreted with caution, as the rates of suicide attempts did not differ significantly across treatment groups. Subsequently, a controlled trial of a single high dose of buprenorphine (32, 64, 96 mg) found that this treatment was associated with a significant reduction in suicidal ideation in patients with comorbid major depression and opioid dependence; all three doses were equally effective in terms of the primary outcome, and no suicide attempts were reported [54]. In this trial, the effect of a single dose appeared to be maintained when patients were followed up after two weeks; however, the lack of a placebo group in this study was a significant limitation. These results suggest that μ-opioid receptor agonism is a potentially valid strategy fora reduction in suicidal ideation, though its effect on suicidal behaviour remains uncertain. There is no evidence for any long-term effect of this approach on suicidality; in an open-label trial of combined buprenorphine/samidorphan in patients with depression, no significant effect on suicidal ideation was observed at 6- and 12-month follow-ups, suggesting that the effects of this drug on suicidality may be confined to the acute phase of treatment [61]. Further randomized control trials of buprenorphine or related agents with a focus on outcomes beyond suicidal ideation are required, both to establish its real-world efficacy and to determine the effective dose range and the risk/benefit ratio of this approach.

The effects of buprenorphine on suicidal ideation or behaviour may not be confined to its μ-opioid receptor partial agonism: there is some evidence that a blockade of κ-opioid receptors may also attenuate suicide risk, and that this property contributes to the putative "anti-suicidal" effect of buprenorphine [56]. In animal models, δ-opioid agonists also reduce the affective component of experimentally induced pain [59]. These receptors may represent useful alternate targets for the pharmacological reduction in suicide risk through the attenuation of social pain.

(b)   Serotonin receptor agonists

The monoamine transmitter serotonin has been hypothesized to play a key role in the neurobiology of suicidal behaviour. Low levels of the serotonin metabolite, 5-hydroxyindole-acetic acid (5-HIAA), have been consistently reported in suicide attempters [18], and variations in serotonin-related genes, such as the serotonin transporter gene (*SLC6A4*) and the tryptophan hydroxylase-2 gene (*TPH2*) have been associated with suicide attempts [108]. Serotonin is also hypothesized to play a role in the pathogenesis of depression, and drugs acting on the serotonergic pathways are the most commonly used antidepressants [109]; however, a trial of the antidepressant drug duloxetine, which acts by inhibiting the reuptake of both serotonin and norepinephrine, found that this drug did not reduce social pain when administered over a period of 1 month to patients with fibromyalgia [65]. This lack of effect on social pain may partly explain why conventional antidepressants have mixed or inconsistent effects on suicidality [110,111].

Recent evidence suggests that the serotonergic system plays a role in the modulation of social pain. A functional polymorphism of the *SLC6A4* gene is associated with the perception of interpersonal problems in depressed individuals [112], and the administration of an agonist of serotonin type 2A and 1A receptors appears to reduce social pain processing [39]. *d*-fenfluramine, which increases serotonin release, is associated with reduced distress when performing a socially stressful task, as is the serotonin type 1A partial agonist ipsapirone. On the other hand, the stimulation of serotonin type 1B receptors, or blockade of type 2A

receptors, appeared to increase anxiety during such tasks [113]; thus, it is possible that pharmacological therapies which selectively target specific types of serotonin receptors can reduce the perception of social pain.

A promising lead for a serotonergic drug that might reduce suicidal ideation or behaviour was obtained from epidemiological studies of individuals taking psychedelic and related drugs for recreational purposes. These drugs, which include psilocybin, lysergic acid diethylamide (LSD), dimethyltryptamine and mescaline, all act either through an increase in presynaptic serotonin release or through the agonism of serotonin type 1 and type 2 receptor subtypes. Initially, a class effect was observed for these drugs, with lifetime psychedelic use associated with lower rates of suicidal ideation, planning, and attempts [63,66]. Further research led to the emergence of a more fine-grained picture, in which LSD was associated with an increase in the risk of depression and suicidal ideation, while psilocybin was associated with reduced suicidal ideation and behaviour [67,68,72]. Psilocybin was also found to reduce social pain in a controlled trial conducted in healthy volunteers [39]. Subsequently, the use of a single dose of psilocybin, in combination with psychotherapy, was noted to cause a significant reduction in suicidal ideation in patients with advanced cancer. This effect was sustained when the patients were followed up 6 months later [71]. These results suggest that psilocybin may be an effective means of targeting the social pain pathway and causing an acute reduction in suicide risk; however, in a recent randomized trial involving patients with depression, the combination of psilocybin and psychotherapy did not differ from waiting-list controls in terms of the levels of suicidal ideation [64]. In addition, caution is required when using psilocybin in a clinical setting, as it has also been associated with undesirable psychological effects ("bad trips") such as anxiety, aggression and dysphoria [114].

3,4-methylenedioxymethamphetamine (MDMA), commonly referred to as "Ecstasy", is a drug that is sometimes grouped together with psychedelics or hallucinogens because of an overlap in their pharmacological properties and perceived effects The mechanism of action of MDMA appears to involve an increase in the release of serotonin, though it also has effects on dopamine and noradrenaline release [115]. MDMA, like psilocybin, appears to be associated with reduced suicidal ideation in community surveys [68] Recently, MDMA has been investigated as a potential treatment for patients with post-traumatic stress disorder in combination with psychotherapy; however, the results of the two trials published to date are inconsistent, with one trial reporting a significant reduction in suicide risk at study completion [70], and the other reporting no difference between MDMA and a placebo in terms of suicidality [71]. In the trial reporting positive outcomes, a transient increase in suicidal ideation was observed during the initial sessions of MDMA-assisted psychotherapy at certain doses [70]. Given the numerous reports of suicide attempts or suicidality associated with MDMA outside a clinical context, as well as consistent reports of neurotoxicity and other end-organ damage associated with excessive MDMA use [115–117], the evaluation of MDMA for the purpose of suicide prevention requires careful supervision and analysis of the risk–benefit ratios.

(c)    Glutamate receptor antagonists

Glutamate plays a key role in pain perception, and functional imaging studies have demonstrated elevated levels of glutamate in patients with chronic pain syndromes [118–121]. The effects of glutamate on pain are mediated through its actions on specific receptors, including the *N*-methyl d-aspartate (NMDA) receptor and metabotropic receptors [122,123]. A pharmacological blockade of the NMDA receptor reduces central pain perception, and this effect appears to be mediated by the actions of endogenous opioids at μ receptors [123,124]. In rodent models, glutamate plays a key role in the response to social stressors, such as exclusion or exposure to hostile behaviours, and this effect appears to be related to alterations in the activity of NMDA and metabotropic type 5 (mGlu5) glutamate receptors [79,82]. These findings suggest a plausible association between glutamatergic transmission and neural mechanisms related to social pain. A study of adolescents with suicidal ideation found evidence of altered glutamate metabolism in the ACC, a region

strongly associated with the experience of social pain [83]. Based on these findings, it has been suggested that glutamate-related therapies may represent a novel approach to the reduction in suicidality [125].

A short-term clinical trial of pregabalin, a drug that inhibits the release of glutamate, found that this drug did not reduce social pain in patients with fibromyalgia, despite producing significant improvements in physical pain [65]. On the other hand, the NMDA receptor antagonist ketamine, previously used as a general anaesthetic and analgesic, has recently been approved for use in patients with resistant depression. Apart from its rapid antidepressant properties, ketamine appears to exert a significant early effect on suicidal ideation in patients with mood or anxiety disorders [86]. Oral ketamine was found to exert a beneficial effect on suicidal ideation in the case of a patient with resistant depression and chronic suicidality [76]. A small open-label trial of oral ketamine, involving patients in hospice care with depression, found evidence of a beneficial effect for depressive symptoms, but no specific effect on suicidality [81]. A subsequent clinical trial of oral ketamine, involving 32 patients with a history of chronic suicidal thoughts found that ketamine was associated with a reduction in suicidal ideation over a 6-week period, with over two-thirds of participants reporting a significant benefit. These effects were independent of the patients' formal diagnosis or concurrent treatment with other medications, and were associated with increased grey matter volumes in brain regions that form part of the "social pain circuit", such as the thalamus and periaqueductal gray [73,77]. It has also been observed that the anti-suicidal effects of intravenous ketamine in humans are blocked by the administration of naltrexone, a μ-opioid receptor antagonist [85]. A synthesis of these results is consistent with the hypothesis that ketamine may reduce suicidal ideation through its effects on social or psychological pain, mediated by increased activation of the μ-opioid receptor; however, the specific effects of ketamine on suicidal behaviour, as distinct from ideation, have not been evaluated independently.

More recently, trials of intranasal esketamine (S(+) ketamine), an enantiomer of ketamine, have been conducted with a specific focus on suicidal ideation and behaviour; however, the results of these trials have been mixed, for example, esketamine was found to be consistently superior to the placebo for depressive symptoms [74,78,80], but only one trial identified a meaningful difference between esketamine and the placebo for suicidal ideation [74]. A head-to-head comparison of ketamine and esketamine, both given as a single intravenous infusion, found that both drugs were of comparable efficacy in reducing suicidal ideation in patients with treatment-resistant depression, and this effect was sustained at 7 days post-infusion [84]. These results suggest that the efficacy of esketamine on suicidality may be influenced by the route of administration.

Given the known risks of misuse associated with ketamine [126], the polyamine molecule agmatine has also been investigated as a safer alternative. Agmatine appears to act at least partially through a blockade of the NMDA receptor [127], and a post-mortem study has documented reduced agmatine levels in the cerebral cortex of completed suicides, independent of the presence of depression [75]. Agmatine has also been shown to have antidepressant-like effects in animal models of depression [128], and it could theoretically share anti-suicidal effects with ketamine; however, these properties have not yet been evaluated in controlled trials in humans.

(d)    xytocin receptor agonists

The neuropeptide oxytocin functions both as a hormone and a neurotransmitter, and plays a key role in the regulation of social behaviour in animals as well as humans. These effects are mediated through the oxytocin receptor (OXTR), which is present in high density at several regions of the brain, including the cerebral cortex, hippocampus, and nucleus accumbens [129]. Oxytocin and endogenous opioid peptides appear to interact with each other in a reciprocal manner and oxytocin reduces pain perception through effects that are mediated by increased opioid receptor activation [96]. In rodents, oxytocin receptor levels decrease with social isolation, and the administration of oxytocin partially reduces the behavioural problems seen in socially isolated mice or rats [94]. In healthy

human volunteers, oxytocin reduces the social pain caused by romantic rejection [98], and a functional polymorphism of the *OXTR* gene influences changes in HPA axis functioning in individuals exposed to social rejection [87]. In individuals who had made a suicide attempt, exposure to social rejection in an experimental setting resulted in a decrease in plasma oxytocin levels, and the serum oxytocin and cerebrospinal fluid (CSF) levels of oxytocin were lower than in the controls. Additionally, CSF oxytocin levels were negatively correlated with the intent associated with a suicide attempt [88,92,93]. These findings are consistent with a significant relationship between oxytocin and suicidal behaviour which may be mediated, at least in part, by the effects of oxytocin on social pain; however, it should not be assumed that a simple, negative linear relationship exists between oxytocin and suicidality. For example, in adolescents with high levels of anxiety, negative interactions with peers were associated with suicidal ideation in those with a high level of oxytocin [95].

The intranasal administration of oxytocin has been evaluated as a therapeutic option in several psychiatric disorders, including depression and anxiety; however, evidence of efficacy has been inconclusive to date [89]. Moreover, studies examining the specific effects of oxytocin on the response to experimentally induced social pain have yielded negative results in direct analyses [90,91,97]. The beneficial effects of oxytocin on social pain may be confined to specific subgroups, such as women [91] and those without an avoidant attachment style [90]. Besides these individual factors, methodological limitations related to sample size, study design, treatment duration, and the definition of specific outcomes may account for the inconsistent responses seen in clinical trials of intranasal oxytocin [130]. Given its favourable adverse effect profile, and its lack of potential for misuse compared to many of the drugs discussed above, well-designed short-term clinical trials of oxytocin in individuals with suicidality related to social pain may still be warranted.

(e)    Other therapeutic targets

Research on the neural and molecular mechanisms of social pain has identified certain other approaches that may reduce social pain in an experimental setting, though agents targeting these pathways have not yet been evaluated in controlled clinical trials. These potential targets are summarized briefly in this section.

Suicidality is associated with significant alterations in the levels of several cytokines, including elevations in interleukin-6 (IL-6) and interleukin-1 beta (IL-1β) [20]. At a central level, the upregulation of tumor necrosis factor alpha (TNF-α) has been demonstrated in the prefrontal cortex of suicide victims [131]. Changes in the levels of these inflammatory markers are also significantly associated with social pain. In individuals exposed to social rejection in a laboratory setting, increased levels of IL-6 and soluble tumour necrosis factor-alpha receptor type II (sTNFαRII) were observed, and these changes were correlated with increased activity in brain regions linked to social pain [106]. In a similar study involving only female participants, exposure to criticism in an experimental setting was associated with increases in brain amygdala activity that correlated with increases in IL-6 levels and self-reported social feelings of rejection [103]. The induction of an inflammatory response through the administration of low-dose endotoxin was also associated with elevated IL-6, depressed mood, and social pain when healthy individuals were exposed to social rejection in an experimental setting. These changes were correlated with alterations in brain regions hypothesized to be related to social pain, such as the dorsal ACC and anterior insula, particularly in female participants [42]. These findings are suggestive of an overlap between the inflammatory correlates of social pain and the changes seen in suicide attempters.

It is possible that, in patients with a suicide attempt occurring in relation to social pain, and in association with elevated levels of these markers, immunomodulatory or anti-inflammatory therapies may reduce suicidal ideation or behaviour through their effects on social pain. In a trial involving patients with resistant depression, the TNF-α antagonist infliximab was associated with a significant reduction in suicidality compared to a placebo over a period of 12 weeks only in patients with elevated baseline C-reactive protein, suggesting that baseline inflammatory activity is an important predictor of response to

immune therapies [105]. However, given the complex and homoeostatic role often played by cytokines, such therapies may not always work in a linear manner [132].

The endogenous cannabinoid or endocannabinoid system has been identified as playing a significant role in several psychiatric disorders, as well as in suicidal behaviour [133]. These effects appear to be mediated primarily through the cannabinoid type 1 ($CB_1$) receptor, which is expressed at high levels in several key brain regions, including the prefrontal cortex and limbic system [134]. In a rat model, changes in social behaviour and a sensitivity to physical or social pain caused by rejection were associated with an up-regulation of $CB_1$ receptors. These changes could be reversed by the administration of rimonabant, a $CB_1$ receptor inverse agonist [41]. The administration of another $CB_1$ inverse agonist, AM-251, was associated with antidepressant effects in a mouse model, and these effects were potentiated by the opioid antagonist naltrexone, suggesting a link between opioid system activity and these effects [104]. Post-mortem studies have found evidence of increased cannabinoid receptor density and functional coupling in the brains of individuals with depression who committed suicide [101,102]. The endocannabinoid system also interacts significantly with other molecular pathways implicated in social pain, such as glutamatergic neurotransmission and the HPA axis [135]; therefore, the endocannabinoid system may be implicated in social pain as well as in suicide. In both these cases, the $CB_1$ receptor appears to be implicated.

However, the administration of rimonabant in humans is associated with a paradoxical increase in depression, anxiety and even suicide risk; these effects were severe enough to necessitate its withdrawal from several markets. These adverse effects may reflect genetic variations in the CB1 receptor gene, or the "inverse agonist" pharmacological properties of rimonabant. Such effects could be minimized or even avoided entirely through the development of pure antagonists, partial agonists or allosteric modulators of the $CB_1$ receptor, but there is as yet no cannabinoid receptor modulator of this sort that is safe or available for clinical use [136].

Acetaminophen (paracetamol), a commonly used analgesic and antipyretic, has been shown to reduce social pain in response to rejection in experimental settings [100], and was also found to reduce social pain in a 3-week clinical trial [107]; however, in the latter trial, acetaminophen was superior to the placebo only in those subjects who reported higher levels of forgiveness. In a study designed to examine whether acetaminophen could increase the capacity for suicide in healthy volunteers, there was no evidence that this drug was associated with either an increase or a decrease in suicidal capacity [99]. Given the frequent use of acetaminophen in intentional overdoses [137], this drug cannot be readily recommended as a potential therapeutic option for suicidal ideation.

## 4. Discussion

Suicide often appears to be the result of intolerable psychological pain, and such pain often arises in the context of social rejection, exclusion, isolation, or the disruption of established social bonds. Though it cannot account for all cases of completed suicide, the social pain model is a useful heuristic both in terms of understanding the contributions of psychosocial factors related to suicide, and in terms of identifying potential targets for intervention. The evidence covered in the current review suggests that a number of potentially useful pharmacological approaches to suicide prevention, particularly in individuals with suicidal ideation or a recent suicide attempt, can be derived from this hypothesis. In many cases, this evidence is limited and inferential in nature, and has not yet been subjected to rigorous evaluation in the actual populations of interest. Nevertheless, it is important to note that at least two of the recently advocated treatments for the rapid reduction of depression and suicidality—ketamine and psilocybin—may exert their effects, at least in part, through a reduction in social pain. Positive results obtained in controlled clinical trials of buprenorphine, though they cannot be generalized, provide further support for the potential value of targeting the "social pain pathway" as an approach to secondary suicide prevention.

Much remains unknown about the "anti-suicidal" properties of these drugs. Are they achieved after a single dose, or after repeated dosing? How long are these effects sustained? Do reductions in suicidal ideation translate into meaningful reductions in suicidal attempts or deaths? What are the potential interactions between these drugs and other treatments that a suicidal patient might receive, such as antidepressants? Are there any long-term adverse effects of these treatments that may not be evident in short-term trials? A definitive answer to these questions can be obtained only through carefully designed controlled clinical trials [138]. The essential components of these trials would be: (a) the inclusion of subjects with a recent suicide attempt and active suicidal ideation occurring in the context of social pain, regardless of the presence of a formal psychiatric diagnosis such as "major depression" or "anxiety disorder"; (b) the operationalization and measurement of the key variables of interest—social pain, suicidal ideation and suicidal behaviour— before and after treatment; (c) the administration of specific doses of a given agent, either singly or repetitively; (d) comparisons with "treatment as usual", given the paucity of established pharmacological strategies for suicide prevention; (e) a reasonable duration of follow-up, to ensure that any observed effect is sustained over weeks or months and is not a transient and naturally-occurring fluctuation; (f) appropriate ethical safeguards to handle unexpected or paradoxical medication effects and to protect individual participants. The measurement of surrogate markers, such as changes in peripheral blood markers or the activity of specific brain regions, may help in confirming mechanistic hypotheses, but should not be considered a primary outcome in trials of this sort. If specific agents are found efficacious in trials of this sort, they would merit further evaluation in more naturalistic or "real-world" settings. It should also be noted that, given the rarity of suicide attempts in comparison with suicidal ideation, an accurate evaluation of this particular outcome would require fairly large samples and a reasonable follow-up period.

Several precautions related to the use of these drugs, related to their potential for misuse, abuse or unwanted behavioural effects, should be mentioned in this context. These are summarized in Table 2. A careful evaluation of the risk–benefit ratio should precede the initiation of any of these therapies, even if they are found effective, and practitioners prescribing these drugs should take appropriate precautions to ensure that these drugs are not diverted for misuse or prescribed in patients with significant contraindications to their use.

**Table 2.** Limitations and safety concerns associated with novel pharmacological strategies for suicide prevention.

| Concern | Drug | Cause of Mechanism | Risk Reduction Strategy |
|---|---|---|---|
| Lack of evidence | All except ketamine and possibly psilocybin | Lack of controlled clinical trials; variations in definition of suicidal phenotype | Well-designed phase II and III trials with clear operational definitions of suicide-related outcomes |
| Abuse or dependence | Buprenorphine, ketamine, psilocybin, MDMA [114–117,126,139] | Activation of mesolimbic dopamine and opioid pathways; toxicity from accidental or intentional overdoses | Careful selection of subjects for treatment; limited dispensing; screening for misuse at follow-up; legislation. |
| Behavioural toxicity | MDMA, psilocybin, possibly acetaminophen [114–116,140] | Challenging experiences ("bad trips") leading to negative emotional responses and suicidality; blunting of social pain pathway | Careful monitoring of mental status; use of alternate agents; education of patients and caregivers regarding potential adverse effects. |
| Variable efficacy | All | Pharmacogenomic variations in molecular targets or downstream mediators | Genetic analysis of data from controlled trials; pharmacogenomic testing and personalized medicine |

Abbreviations: MDMA, 3,4-methylenedioxymethamphetamine.

Even in the absence of adverse effects, there may be significant variations in the efficacy of these drugs across patients. In some cases, this may reflect the influence of pharmacogenomic factors, such as functional polymorphisms in target receptors or downstream molecular pathways. Secondary analyses of trial data may help in identifying these variants (such as *OPRM1* rs1799971 or *OXTR* rs53576) and lead to the development of pharmacogenomic testing, which would facilitate a personalized medicine-based approach

to the use of these agents [141]. Alternately, the measurement of specific biomarkers may help in identifying subgroups of suicidal patients who would respond to specific therapies, as is already being attempted in the case of depression [142].

Certain limitations should be noted with respect to the current review. First, due to its reliance on information from a variety of sources—translational studies, clinical observation and controlled trials—there is significant heterogeneity in the data presented, and many of the suggestions offered are inferential and tentative in nature. Second, due to variations in the measures of suicidal ideation and behaviour across controlled clinical trials, comparisons of the relative efficacy of different agents are difficult. Third, results obtained in healthy volunteers may have limited validity in patients with an associated psychiatric disorder or those exposed to chronic or severe social stress. Fourth, as this review covered a specific number of databases, it is possible that certain relevant studies may have been omitted inadvertently. Fifth, the social pain model of suicide itself is only a theoretical model, and its applicability to specific cases of suicidal ideation or behaviour requires further empirical validation [30,31].

Finally, it should be noted that these drugs—even if found effective in the reduction of suicidal ideation and behaviour—should not be viewed as "cures" for suicidality. As mentioned earlier, suicide is a complex phenomenon resulting from the interplay of biological, psychological and social factors. It is naïve to assume that a single drug can address the deeper psychological or social roots of suicidal behaviour, even if these have identifiable biological substrates. These drugs should best be viewed as short-term, acute-phase interventions, in the same way as analgesics are used for the management of acute or post-operative pain [143]. They are probably best used as part of a multidisciplinary treatment approach that includes psychological or social interventions tailored to an individual's needs [144].

## 5. Conclusions

Despite certain limitations in the existing evidence base, the social pain hypothesis holds promise for the development of novel suicide prevention strategies, particularly in the immediate aftermath of a suicide attempt or in individuals with active suicidal ideation related to social stress. It is hoped that the findings summarized in this review, though requiring replication and extension, will be of use to those involved in the development and testing of pharmacological therapies for this indication. These therapies may prove to be effective in a short-term reduction in suicidal ideation and behaviour, but should be evaluated carefully for both their efficacy and safety prior to their use in clinical settings.

**Funding:** The work presented in this paper received no funding.

**Institutional Review Board Statement:** Not applicable.

**Informed Consent Statement:** Not applicable.

**Data Availability Statement:** Data sharing not applicable.

**Conflicts of Interest:** The author declares no conflict of interest.

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
