# Peer review of "Pharmacological Strategies for Suicide Prevention Based on the Social Pain Model: A Scoping Review"

_psych, doi:10.3390/psych4030038_

Round 1

Reviewer 1 Report

I admire the author's great insight and inspiration.

The claims of this review contain many very interesting perspectives.

This reviewer can also appreciate the comprehensive pharmacological aspect in this manuscript.

However, the previous findings which were indicated in this manuscript are insufficient for the determination of validation of author's statements.

 necessary for individual consideration are only partial introductions, and 

Even if, at first glance, it seems that only the findings that are convenient for the author are selected. In other word, the reviewer also knows other findings that are the opposite finding against author's opinion in a number of psychopharmacological studies.

This reviewer concerns following these two points of the major reason of this problem.

The major reason is that the pharmacological category targeted by the author is too broad.

The search method is possibly inadequate. 

If author agrees with the reviewer's comments and the author revises significant improvements, the reviewer will be happy to re-review.

Author Response

Responses to review comments and details of corrections made:

1. "I admire the author's great insight and inspiration.

The claims of this review contain many very interesting perspectives.

This reviewer can also appreciate the comprehensive pharmacological aspect in this manuscript."

Response: I thank the reviewer for these positive and encouraging comments on the initial version of the manuscript.

2. "However, the previous findings which were indicated in this manuscript are insufficient for the determination of validation of author's statements."

Response: I agree with this comment by the reviewer. In line with this suggestion, I have revised the entire manuscript substantially to include more original studies (positive and negative) and to remove unnecessary references to commentaries, opinion articles, etc. which do not provide an evidence base. Details of these changes are provided below.

3. "Even if, at first glance, it seems that only the findings that are convenient for the author are selected. In other word, the reviewer also knows other findings that are the opposite finding against author's opinion in a number of psychopharmacological studies."

Response: I agree with this critique by the reviewer. In line with this, the literature has been searched again and the following negative or equivocal clinical trials have also been included in the review: Thase et al., 2020 (reference no. 61), Davis et al., 2021 (reference no. 64), Mitchell et al., 2021 (reference no. 69), Fu et al., 2020 (reference no. 78), Ionescu et al., 2021 (reference no. 80), Irwin et al., 2013 (reference no. 81), Vieira et al., 2021 (reference no. 84), Lebowitz et al., 2019 (reference no. 95). Also, the possibility of a paradoxical initial increase in suicidality has been added to the manuscript as this was reported in Mithoefer et al., 2018 (reference no. 70), and the fact that positive responses were observed only in sub-groups but not in the entire sample was added to the description of Raison et al., 2013 (reference no. 106) and Slavich et al., 2019 (reference no. 108). In each of the relevant sections of the text, these negative or contradictory findings have been summarized for each group of drugs. Full details of all included articles have also been provided in a new Table (Table 1).

A new paragraph on "Limitations of the current review" has also been added as follows:

"Certain limitations should be noted with respect to the current review. First, due to its reliance on information from a variety of sources – translational studies, clinical observation and controlled trials – there is significant heterogeneity in the data presented, and many of the suggestions offered are inferential and tentative in nature. Second, due to variations in the measures of suicidal ideation and behaviour across controlled clinical trials, comparisons of the relative efficacy of different agents are difficult. Third, results obtained in healthy volunteers may have limited validity in patients with an associated psychiatric disorder or those exposed to chronic or severe social stress. Fourth, as this review covered a specific number of databases, it is possible that certain relevant studies may have been omitted inadvertently. Fifth, the social pain model of suicide itself is only a theoretical model, and its applicability to specific cases of suicidal ideation or behaviour requires further empirical validation."

4. "The major reason is that the pharmacological category targeted by the author is too broad."

Response: I agree with this comment by the reviewer. The review has been shortened substantially by the removal of sections on cannabinoids and immune modulators for which there is insufficient evidence. Instead, only drugs for which there are controlled trials in humans have been given a separate section, and the others have been summarized briefly in "Other pathways of interest" without giving them undue weightage. Superfluous sections on psychotherapies, drugs not related to the review topic (antidepressants, lithium, clozapine, etc.) and other speculative aspects have been deleted in line with the reviewer's suggestions.

5. "The search method is possibly inadequate."

I agree with this criticism by the reviewer. A fresh literature search has been conducted. The Methodology section has been rewritten to reflect this (including the coverage of additional search terms and a description of articles included / excluded at each stage) and a new diagram (PRISMA-ScR Flow Diagram, Figure 2) has been provided to address this error in the original manuscript.

Reviewer 2 Report

This is a well-written, interesting literature review on the relationship between the social pain hypothesis and psychopharmacological treatment for the prevention of suicidality. The author reviews the literature on such treatments in light of the social pain hypothesis, with the aim to identify possible prevention strategies. Although the paper has many merits, there are some issues the author needs to address:

- It would be useful for the reader to know what the 'social pain hypothesis' entails while reading the introduction of the paper. Currently, the author only states "In this paper, evidence for one such hypothesis – the social pain model of suicide – is critically evaluated using the available evidence" on page 2, but does not elaborate on what this model entails. 

- The methodology lacks detail. Please refer to the PRISMA 2020 checklist and guidelines (https://prisma-statement.org/ )  in the reporting of both the methodology and the results of your findings. For example, it is unclear what the exclusion and inclusion criteria were, what the exact search strings were, etc. It is also not quite clear to me whether the list of headings on page 3 resulted from the review process, or was pre-determined by the author. 

- Similarly, it is unclear which papers were included in the review - usually a table is provided with more detailed information on each of the included papers, but that is lacking here. Please provide such an overview.

- The author states that the aims of the research are (page 2): "In this paper, evidence for one such hypothesis – the social pain model of suicide – is critically evaluated using the available evidence. Next, potential targets for pharmacological intervention, based on our existing knowledge of the phenomenon of social pain, are identified, and their potential benefits and risks are explored." --> I wonder whether these aims were met. Specifically, I don't know whether the author critically evaluates the social pain hypothesis or merely summarizes the literature that links social pain with suicidality. 

Author Response

1. "This is a well-written, interesting literature review on the relationship between the social pain hypothesis and psychopharmacological treatment for the prevention of suicidality. The author reviews the literature on such treatments in light of the social pain hypothesis, with the aim to identify possible prevention strategies. Although the paper has many merits, there are some issues the author needs to address:"

Response: I thank the reviewer for their encouraging and positive comments on the initial manuscript. I concur with the valuable suggestions they have offered for its improvement and have made corrections as described below.

2. "- It would be useful for the reader to know what the 'social pain hypothesis' entails while reading the introduction of the paper. Currently, the author only states "In this paper, evidence for one such hypothesis – the social pain model of suicide – is critically evaluated using the available evidence" on page 2, but does not elaborate on what this model entails."

Response: I apologize for this shortcoming, which arose from poor organization of the original manuscript. In the revised manuscript, the concept of social pain and the development of the social pain model of suicide are described in detail in the Introduction, along with a schematic representation of the key factors involved in this model (Figure 1, revised manuscript) prior to the description of the review methodology and results.

3. "The methodology lacks detail. Please refer to the PRISMA 2020 checklist and guidelines (https://prisma-statement.org/ )  in the reporting of both the methodology and the results of your findings. For example, it is unclear what the exclusion and inclusion criteria were, what the exact search strings were, etc. It is also not quite clear to me whether the list of headings on page 3 resulted from the review process, or was pre-determined by the author."

Response: I agree with this criticism by the reviewer. The methodology section has been rewritten as follows:

"The current paper is a scoping review of the links between social pain and suicide, with a specific focus on potential targets for pharmacological interventions. This review was carried out in accordance with the PRISMA Extension for Scoping Reviews (PRISMA-ScR) guideline and checklist [53]. For the purpose of this review, a literature search of the PubMed / MEDLINE, Scopus and ScienceDirect literature databases was conducted, using the following search terms:

  • (“social pain”, “social pain model”, “social pain hypothesis”, “social rejection” OR “social stress”) along with (“suicide”, “suicide attempt”, “suicide attempter”, “suicidal ideation”, “suicidal behaviour”, “suicide risk”, “suicide prevention” or “suicidality”). This search yielded a total of 287 citations (PubMed – 140, Scopus – 19, ScienceDirect – 128)
  • (“endogenous opioid”, “opioid receptor”, “serotonin”, “serotonergic”, “cannabinoid”, “endocannabinoid", “oxytocin”, “cytokine” OR “interleukin”) along with(“suicide prevention”, “suicide attempt”, “suicide attempter”, “suicidal ideation”, “suicidal ideator”, “suicide risk” or “suicidality”) along with (“social pain”, “social stress” OR “social rejection”). The purpose of this second search was to identify studies examining the known neural substrates of social pain in relation to suicidality. This search yielded a total of 279 citations (PubMed – 133, Scopus – 87, ScienceDirect – 59).

Thus, a total of 566 citations were screened for possible inclusion in the scoping review. After exclusion of duplicate citations (n = 205), the remaining 361 citations were evaluated for inclusion. Based on an examination of the title and abstract, 163 citations were excluded as being unrelated to the subject of this review. The complete texts of the remaining 198 articles were examined for suitability. Articles were included only if they reported original findings, or syntheses of original research (systematic reviews and meta-analyses), examining the links between social pain and suicide, with a specific focus on neurobiological mechanisms or pharmacological therapies. Studies of psychosocial correlates of social pain, psychotherapy trials, general review articles on suicide or social pain, and commentaries or editorials were excluded, as were papers that were unrelated to the subject of this review. Through this process, 52 papers were selected for inclusion. As a final step, the reference lists of the included articles were searched for relevant articles that might have been missed through the above search; five further papers were retrieved through this method. A total of 57 papers were included in the final review [39, 41, 54-108]. The search strategy is summarized in the PRISMA-ScR flow diagram below (Figure 2)."

A new flow diagram (Figure 2, revised manuscript) has been provided according to the PRISMA-ScR guidelines, and details of the study subheadings are provided in Table 1 and in the first part of the Results section.

4. "Similarly, it is unclear which papers were included in the review - usually a table is provided with more detailed information on each of the included papers, but that is lacking here. Please provide such an overview."

Response: I agree with this criticism by the reviewer. A new table (Table 1) has been added to the paper to address this limitation, providing a detailed description of all included studies.

5. "he author states that the aims of the research are (page 2): "In this paper, evidence for one such hypothesis – the social pain model of suicide – is critically evaluated using the available evidence. Next, potential targets for pharmacological intervention, based on our existing knowledge of the phenomenon of social pain, are identified, and their potential benefits and risks are explored." --> I wonder whether these aims were met. Specifically, I don't know whether the author critically evaluates the social pain hypothesis or merely summarizes the literature that links social pain with suicidality."

Response: I agree with this criticism by the reviewer. The revised manuscript includes several negative studies to provide a critical review and to balance the possible bias in the initial manuscript. A list of limitations of the current review has also been added to the Discussion.

Round 2

Reviewer 1 Report

The authors have responded well to my comments. The present form of the paper is acceptable.